# OpenReview forum: "MJ-Bench: Is Your Multimodal Reward Model Really a Good Judge for Text-to-Image Generation?"
_NeurIPS.cc/2025/Datasets_and_Benchmarks_Track — NeurIPS 2025 Datasets and Benchmarks Track poster_

### Official Review · Reviewer_QC8n · 2025-06-29

**Rating:** 4
**Confidence:** 4

**Summary:**

The paper organizes a new preference dataset called MJ-Bench, which captures four different aspects of text-to-image generation: text-image alignment, safety, image quality, bias, composition, and visualization. The paper also conducts detailed analysis, such as the consistency of the preference of the judges w.r.t different image modes.

The benchmark assesses various judge types, including small CLIP-based scorers, open-source vision-language models (VLMs), and closed-source VLMs.

**Dataset Code Accessibility:**

Yes

**Ethical Considerations:**

No, there are no or only very minor ethics concerns

**Final Justification:**

I choose to keep the positive score.

**Limitations Weaknesses:**

The novelty of this paper appears somewhat limited, as the majority of the pipeline components have been previously explored in existing studies. To strengthen the contribution, the authors are encouraged to clearly highlight the distinctions between their approach and prior work. Additionally, some comparative analyses currently placed in the appendix—such as Table 4—would be more impactful if moved into the main body of the paper to better support the claimed improvements.

**Strengths Contributions:**

1. MJ-BENCH covers a wide range of evaluation perspectives critical to text-to-image generation, offering granular insights.

2. The benchmark systematically compares closed- and open-source models as well as lightweight scorers, providing a nuanced understanding of trade-offs.

3. Extensive experiments, including human-in-the-loop evaluations, lend strong empirical credibility to the conclusions.

4. The methodology, experiments, and evaluations are thoroughly conducted, contributing to the overall clarity and credibility of the work.

---

> ### Author Rebuttal · Authors · 2025-07-31
>
> Dear Reviewer QC8n,
>
> We sincerely thank you for your generous recognition of our work—especially for highlighting the breadth and depth of MJ-BENCH, its systematic comparison across open- and closed-source models, and the rigor of our experimental design. Your appreciation of the benchmark’s granularity, empirical robustness, and methodological clarity means a great deal to us. In the following, we have responded to your insightful comments, and we hope our replies are both clear and constructive in addressing your valuable feedback.
>
>
> > The novelty of this paper appears somewhat limited, as the majority of the pipeline components have been previously explored in existing studies. To strengthen the contribution, the authors are encouraged to clearly highlight the distinctions between their approach and prior work. Additionally, some comparative analyses currently placed in the appendix—such as Table 4—would be more impactful if moved into the main body of the paper to better support the claimed improvements.
>
>
> Thank you for your valuable feedback. We acknowledge the reviewer’s concern regarding the novelty of our work and appreciate the opportunity to clarify our contributions. While some components in MJ-Bench build upon existing practices in reward model evaluation, our work introduces several key novelties that substantially extend the current landscape:
> - **Comprehensive and finer-grained evaluation perspectives:** To the best of our knowledge, MJ-Bench is the first benchmark to comprehensively cover a wide range of perspectives across diverse text-to-image tasks, including composition and, notably, visualization as core evaluation dimensions, i.e., two critical aspects often overlooked by prior benchmarks such as [1-3]. Specifically, MJ-Bench not only covers fundamental perspectives such as alignment, safety, and quality, but also incorporates more sophisticated dimensions like visualization (e.g., academic diagrams, OCR quality), a newly emerging capability in advanced models like GPT-4o-Image that has not been systematically assessed before. This richer set of perspectives enables MJ-Bench to uncover failure cases missed by benchmarks that focus solely on alignment or safety.
> - **Comprehensive and systematic evaluation protocol:** MJ-Bench introduces a multi-faceted evaluation design covering a wide range of judge types (CLIP-based, single/multi-input VLMs, open-/closed-source models), input settings (single vs. paired image input), and scoring modes (numeric vs. Likert scales, with vs. without tie threshold). Furthermore, our evaluation goes beyond static preference matching, where we also conduct end-to-end evaluation with such judge feedback in-the-loop (e.g. via RLAIF) during post-training and evaluate their output using human annotators. This two-stage evaluation including offline preference and end-to-end alignment provides a more holistic view of reward model reliability and usefulness in practice.
> - **Advanced reward model training:** In addition to benchmarking existing judges, we further demonstrate the utility of MJ-Bench by training our own reward model on the dataset as shown in Table. 39 in appendix C.7. This experiment illustrates MJ-BENCH’s value not only as an evaluation suite, but also as a training resource for developing improved reward models for guiding advanced multimodal alignment for high-quality text-to-image generation.
>
> Regarding the suggestion to move Table 4 (dataset comparison) into the main body, we fully agree. In our revised version, we will move it to the main text to better highlight the improvements of MJ-Bench over prior datasets such as HPD [1-2], Pick-a-Pic [3], ImageRewardDB [4],  and VisionPrefer [5].
>
>
> We hope that these clarifications better articulate the novelty and practical contributions of MJ-Bench. We sincerely appreciate your constructive suggestions, which have helped us improve the presentation of our work. If our response have addressed your concerns, we would be truly grateful if you could consider raising your score to help us share MJ-Bench with a broader research community.
>
> Please don’t hesitate to let us know if anything remains unclear or if further clarification would be helpful. Thank you again for your time and thoughtful feedback!
>
>
> Best regards,
>
> Submission #884 Authors
>
> **Reference**
>
> [1] Wu, Xiaoshi, et al. "Human preference score: Better aligning text-to-image models with human preference." Proceedings of the IEE/CVF International Conference on Computer Vision. 2023. \
> [2] Wu, Xiaoshi, et al. "Human preference score v2: A solid benchmark for evaluating human preferences of text-to-image synthesis." arXiv preprint arXiv:2306.09341 (2023). \
> [3] Kirstain, Yuval, et al. "Pick-a-pic: An open dataset of user preferences for text-to-image generation." Advances in neural information processing systems 36 (2023): 36652-36663. \
> [4] Xu, Jiazheng, et al. "Imagereward: Learning and evaluating human preferences for text-to-image generation." Advances in Neural Information Processing Systems 36 (2023): 15903-15935. \
> [5] Wu, Xun, Shaohan Huang, and Furu Wei. "Multimodal large language model is a human-aligned annotator for text-to-image generation." arXiv preprint arXiv:2404.15100 (2024).

---

### Official Review · Reviewer_eVhG · 2025-07-01

**Rating:** 4
**Confidence:** 4

**Summary:**

This paper introduces MJ-BENCH, a comprehensive benchmark designed to evaluate the capabilities of multimodal models as "judges" for text-to-image generation. The authors construct a high-quality preference dataset spanning six key dimensions: alignment, safety, image quality, bias, composition, and visualization, each broken down into granular sub-categories. The study evaluates a wide range of judges, from smaller CLIP-based scoring models to large-scale open-source and closed-source VLMs. A key finding is that while closed-source VLMs like GPT-4o generally perform best, smaller scoring models can excel in specific areas like alignment and quality. The practical utility of the benchmark is demonstrated by using the judges' feedback to fine-tune a Stable Diffusion v1.5 model via RLAIF, with human evaluations confirming that the benchmark's automatic metrics align well with end-user preferences.

**Additional Feedback:**

- Could the authors provide a more detailed qualitative and quantitative analysis of common failure modes for different classes of judges? Showing specific examples where, for instance, GPT-4o succeeds but open-source models fail would offer deeper insights into their underlying reasoning capabilities and limitations.
- Have the authors considered validating the benchmark's utility on at least one other major generative architecture (e.g., SDXL)? This would significantly strengthen the claim that MJ-BENCH is a general-purpose tool for improving text-to-image models.
- How do the proposed bias metrics relate to each other and to real-world, perceptible biases? Could a model score well on these metrics yet still exhibit harmful stereotypical associations?

**Dataset Code Accessibility:**

Yes

**Dataset Code Comments:**

The data can be downloaded from Huggingface, and the data loading and model training code can be found on Github.

**Ethical Comments:**

It is evident that there are no ethical concerns.

**Ethical Considerations:**

No, there are no or only very minor ethics concerns

**Final Justification:**

After reviewing the authors’ response and the other reviewers’ comments, I’m satisfied that my concerns have been addressed. I’m keeping my original score and support accepting the paper.

**Limitations Weaknesses:**

- The paper presents results primarily as aggregate scores across high-level categories (Table 2). While useful for a general overview, this misses a crucial opportunity to analyze why and where specific models fail. For example, under the "Composition" dimension, which physical laws are most frequently misunderstood by judges? In "Alignment", do judges struggle more with complex spatial relationships or with attribute binding?
- The RLAIF experiments are conducted exclusively on Stable Diffusion v1.5. While a reasonable choice for a proof-of-concept, the T2I landscape is diverse and rapidly evolving, with architectures like SDXL, SD3, and various Transformer-based models (e.g., FLUX) becoming prominent. The effectiveness of a reward signal can be highly dependent on the underlying generative architecture. The conclusions drawn from fine-tuning SD-1.5 may not generalize to these other models.
- The introduction of ACC, GES, and NDS for evaluating bias is novel. However, the paper could benefit from a clearer discussion of their practical interpretation. A model might achieve a high GES (indicating score equality) but still assign consistently low scores to a particular demographic group across all occupations, which is a form of bias in itself.

**Strengths Contributions:**

- The authors create a meticulously structured benchmark. By decomposing the evaluation into six dimensions and numerous sub-scenarios, the authors provide a much-needed framework for a fine-grained analysis of multimodal judges.
- The paper conducts a broad and systematic comparison of various judge models. The analysis of different feedback modalities provides valuable insights into the practical application of these models as reward providers.
- The authors close the loop by using the feedback from these judges to fine-tune a generative model and then subjecting the results to human evaluation. This demonstrates the benchmark's practical relevance and utility for the broader community working on model alignment.

---

> ### Author Rebuttal · Authors · 2025-07-31
>
> Dear Reviewer eVhG,
>
> Thank you so much for your valuable suggestions to help improve our paper. We greatly appreciate your recognition of the novelty and comprehensiveness of our work. Below, we provide detailed, point-by-point responses to your questions, and we hope these address your concerns.
>
> >  analysis of fine-grained subcategory results such as alignment and composition
>
> We appreciate this insightful comment regarding the need for a more fine-grained analysis of model performance within higher-level categories such as composition and alignment.
>
> Regarding subcategory-wise analysis for alignment, we have already provided a comprehensive comparison of each evaluated model on individual alignment subcategories including object, attribute, action, location, and count in Tables 19–22 of Appendix C.6.1, along with detailed analysis. Our findings are consistent with the reviewer’s observations, where we find that judges typically struggle more with complex spatial relationships and action patterns, while performing better on object identification and attribute binding tasks. Additionally, we find that score-based models excel at object and attribute subcategories but tend to underperform relative to VLM judges on spatial and count tasks, which demand more advanced visual reasoning capabilities.
>
> Regarding even more fine-grained dimensions such as those under the _Physical Law_ subcategory, we completely agree that distinguishing specific types of physical laws within the composition dimension could provide additional valuable insight. In response, we conducted further analysis by grouping the dataset samples under physical law into three categories: (1) _Classical Mechanics_ (e.g., Newton’s laws, gravity); (2) _Structural Mechanics_ (e.g., stability, support); and (3) _Optics_ (e.g., shadow, light composition). The results for a subset of models are shown in Table-r.1 below, and we will provide the full list of results in the revised version. Our findings indicate that while models generally perform well on _Optics_ and _Structural Mechanics_, they encounter greater difficulty when judging against _Classical Mechanics_. This suggests that judge models can more easily provide feedback for aspects such as shadowing and object interactions with embedded knowledge, but require more advanced visual perception and reasoning to accurately assess feedback related to classical mechanics such as gravity.
>
>
> _Table-r.1: Performance on physical law subcategories under the composition dimension (best performance in bold)._
>
> | Judge model        | Classical Mechanics | Structural Mechanics | Optics | Average |
> |--------------------|--------------------|---------------------|--------|---------|
> | GPT-4o             | 62.3               | 72.6                | 79.4   | 65.1    |
> | OpenAI-o1          | 69.0               | **91.0**            | **92.5** | **83.0**|
> | Gemini-2.5-Pro     | 59.8               | 89.2                | 87.0   | 78.6    |
> | Claude-3.7-Sonnet  | **70.5**           | 76.2                | 89.6   | 75.4    |
>
> We sincerely appreciate the reviewer’s constructive feedback, which has helped us strengthen the paper. We will be sure to highlight this more systematic, fine-grained analysis in the main text of the revised version.
>
>
> > validating benchmark's utility for RLAIF on other major generative architectures (e.g., SDXL) beyond SD-1.5
>
> We sincerely appreciate the reviewer’s suggestion and fully agree that evaluating on a broader set of state-of-the-art image generation models is important for assessing the generalizability of conclusions drawn from MJ-Bench. In response, we have conducted additional RLAIF experiments using SDXL, comparing the feedback from three representative judge models, i.e., GPT-4o, Internvl-chat-v1-5, and HPS-v2.1. The human evaluation results are presented in Table-r.2 below.
>
> _Table-r.2: Human evaluation results on images generated by SDXL models fine-tuned with feedback from three multimodal judges. Metrics include: ranking over fixed seed (FR), ranking over random seed (RR), average ranking (AR), and average voting (AV). The best performance in each column is bolded._
>
> | Model                | Align-FR | Align-RR | Align-AR | Align-AV  | Safe-FR | Safe-RR | Safe-AR | Safe-AV  | Bias-FR | Bias-RR | Bias-AR | Bias-AV  |
> |----------------------|:--------:|:--------:|:--------:|:---------:|:-------:|:-------:|:-------:|:--------:|:-------:|:-------:|:-------:|:--------:|
> | GPT-4o               | 2.07 | 1.94 | 2.04 | 35.9% | **1.66** | **2.01** | **1.89** | **42.7%** | **2.01** | 2.49    | 2.14 | 34.2%   |
> | Internvl-chat-v1-5   | 2.39 | 2.73 | 2.50 | 23.7% | 1.81      | 1.97     | 1.93    | 31.5%    | **2.01** | 2.87    | 2.63 | 29.5%   |
> | HPS-v2.1             | **1.54** | **1.33** | **1.46** | **40.4%** | 2.53   | 2.02    | 2.18    | 25.8%    | 1.98    | **0.64** | **1.23**| **36.3%**|
>
> Our findings indicate that the performance trends observed for SDXL closely mirror those for SD-1.5 and the preference-based evaluation in MJ-Bench. Specifically, score-based judge models tend to provide more accurate feedback for alignment and bias, while more sophisticated VLMs yield better feedback on safety. This consistency across architectures further supports the generalizability of our conclusions. We plan to extend our evaluation to additional generative models, such as SD3 and transformer-based models like FLUX, in future work.
>
> We hope these additional experiments and analyses address the reviewer’s concerns regarding the applicability of our approach to the broader T2I model landscape.
>
>
> > more discussion on bias-related metrics such as ACC, GES, and NDS and their real-world implications
>
> We sincerely thank the reviewer for acknowledging the novelty of the proposed bias-related metrics, and we have further clarified their practical interpretation and real-world implications as follows.
>
> - **ACC (Accuracy)** quantifies how often a model’s pairwise decisions are consistent with unbiased reference judgments. In practice, a high ACC suggests that the model is not systematically favoring one group over another in direct comparisons, which is important for mitigating overt, instance-level bias.
>
> - **GES (Gini-based Equality Score)** captures how equally scores are distributed across all demographic groups. A high GES reflects a lack of large disparities in scores and is intended to highlight overall fairness. However, as the reviewer insightfully notes, GES does not capture whether all scores are uniformly low (or high) for a particular group. In real-world settings, this could mask “allocative bias”, where a group receives uniformly lower scores, despite score equality within that group. For instance, if all members of a demographic receive similarly low scores for all occupations, GES alone would not reveal this systematic disadvantage.
>
> - **NDS (Normalized Dispersion Score)** measures the consistency of scoring, i.e., whether the model’s scores are concentrated or dispersed around the mean. In practice, high NDS indicates that the model does not arbitrarily vary its scores, which is desirable for stable and predictable decision-making. However, NDS does not by itself indicate whether the central tendency is fair across groups.
>
> It is important to emphasize that these three metrics are intended to be used together, as they compensate for each other’s limitations. For example, in the scenario raised by the reviewer where a group receives consistently low scores across all occupations, artificially inflating the GES, such bias would likely be reflected by both low ACC and low NDS. Therefore, jointly examining ACC, GES, and NDS provides a more comprehensive and reliable assessment of model bias.
>
> Nevertheless, we agree that aggregate metrics may still overlook certain forms of demographic bias. Therefore, it is essential to complement ACC, GES, and NDS with group-specific descriptive statistics and visualizations (such as per-group means, boxplots, or disparity ratios) to identify and diagnose such issues. We plan to follow the reviewer’s suggestion and include these additional analyses in our revised version for a more comprehensive bias evaluation.
>
>
> > analysis of common failure modes for different classes of judges
>
> We appreciate the reviewer’s suggestion and have expanded the paper to provide both qualitative and quantitative analyses of common failure modes for different judge models. Notably, we find that GPT-4o and other closed-source MLLMs are much more reliable than open-source MLLMs and CLIP-based scorers on nuanced safety and bias cases. For example, as detailed in the paper, GPT-4o consistently flags subtle instances of bias and unsafe content that open-source models and CLIP-based models often overlook, reflecting their stronger reasoning and instruction-following capabilities.
>
> Conversely, CLIP-based models can occasionally outperform MLLMs on strict text-image alignment but suffer from high output variance and greater inconsistency, such as changing preferences based on input order. We have included representative error cases and aggregated failure statistics in the revised paper to illustrate these differences, which we believe offer valuable insight into the unique strengths and limitations of each class of judge.
>
>
>
> We hope that these clarifications better articulate the novelty and practical contributions of MJ-Bench. We sincerely appreciate your constructive suggestions, which have helped us improve the presentation of our work. If our response have addressed your concerns, we would be truly grateful if you could consider raising your score to help us share MJ-Bench with a broader research community.
>
> Please don’t hesitate to let us know if anything remains unclear or if further clarification would be helpful.
>
> Thank you again for your time and thoughtful feedback!
>
>
> Best regards,
>
> Submission #884 Authors

---

> > ### Comment · Reviewer_eVhG · 2025-08-05
> >
> > Thank you for the detailed response. After considering the authors' rebuttal and the other reviewers' opinions, I have decided to maintain my original rating and lean towards accepting this paper.

---

> > > ### Author Response · Authors · 2025-08-05
> > >
> > > Dear Reviewer eVhG,
> > >
> > > Thank you for your positive feedback and for recommending acceptance of our paper! We sincerely appreciate your valuable suggestions, which are instrumental in helping us further improve our work. Thank you again for your valuable support!!
> > >
> > > Best regards,
> > >
> > > Submission #884 Authors

---

### Official Review · Reviewer_PPqG · 2025-07-02

**Rating:** 6
**Confidence:** 2

**Summary:**

For applying text-to-image (T2I) diffusion models to real-world problems, some studies proposed to fine-tune them with a VLM as a specific reward function. However, VLMs' inadequate outputs make T2I models undesirable. This paper addressed this problem by proposing a new benchmark composed of six perspectives.

**Dataset Code Accessibility:**

Yes

**Ethical Considerations:**

No, there are no or only very minor ethics concerns

**Final Justification:**

As described in "Response to the authors", all of my concerns are solved. Therefore, I raised my score 5 -> 6.

**Limitations Weaknesses:**

- MJ-BENCH evaluates data across six key perspectives: alignment, safety, image quality, bias, composition, and visualization. Why are these six perspectives employed? Are other perspectives considered?
- In L89, the notation for M uses 'n' for both the upper-right and lower-right subscripts, which may cause ambiguity. It is recommended to revise one of them.
- In L104, the reference to Fig. x is missing.

**Strengths Contributions:**

- The motivation of this work is clear. For applying text-to-image (T2I) diffusion models to real-world problems, correct multimodal judges are crucial.
- The paper provides the analysis related to CLIP, open-source VLMs, and closed-source VLMs. This analysis provides useful insights.

---

> ### Author Rebuttal · Authors · 2025-07-31
>
> Dear Reviewer PPqG,
>
> We are truly grateful for your insightful suggestions, which have greatly contributed to improving our manuscript. We sincerely appreciate your recognition of the contributions of our work. Below, we have carefully prepared a detailed, point-by-point response to your comments, and we hope it effectively addresses your thoughtful concerns.
>
> > MJ-BENCH evaluates data across six key perspectives: alignment, safety, image quality, bias, composition, and visualization. Why are these six perspectives employed? Are other perspectives considered?
>
> Thank you for insightful question. MJ-BENCH evaluates multimodal reward models across six key perspectives—alignment, safety, image quality, bias, composition, and visualization—because these dimensions collectively capture the **most critical and representative** failure modes of current text-to-image generation models.
> - Alignment ensures the generated image accurately reflects the given instruction, addressing the common issue of hallucination.
> - Safety addresses the risk of generating harmful, toxic, or NSFW content, which is crucial for responsible deployment.
> - Image Quality assesses visual fidelity, including artifacts like blurriness, unnatural textures, or anatomical distortions.
> - Bias targets fairness by evaluating if judges exhibit demographic or occupational biases, especially when scoring images depicting different genders, races, or professions.
> - Composition focuses on spatial consistency, physics, and occlusion—important for generating plausible scenes.
> - Visualization evaluates structured content generation (e.g., academic diagrams, charts), which tests higher-order reasoning and layout capabilities.
>
> These perspectives were selected based on **empirical observations of common failure cases in state-of-the-art models (e.g., GPT-4o-image, SDXL)**, supported by prior work[1-5] in safety, alignment, and evaluation benchmarks. We also ensure each perspective is decomposed into fine-grained subcategories, enabling a more interpretable and diagnostic assessment.
>
> We acknowledge that other perspectives—such as creativity or emotion—may also be relevant in certain use cases, but we focused on the six above due to their measurability, relevance to model alignment, and impact on real-world safety and reliability. Future iterations of MJ-BENCH could expand to include these additional dimensions as evaluation methods mature.
>
>
> > In L89, the notation for M uses 'n' for both the upper-right and lower-right subscripts, which may cause ambiguity. It is recommended to revise one of them.
>
> Thank you for pointing this out. We sincerely apologize for any inconvenience this may have caused you. We agree that using 'n' for both the upper-right and lower-right subscripts can be ambiguous. We will revised the notation in the updated version to avoid this confusion. **For example, by changing the upper index to 𝑖 to denote the sample index, while keeping the lower index to indicate the negative sample.** This improves clarity and consistency in our notation. Thank you once again for your valuable feedback—it truly contributes to enhancing the quality of our work
>
> > In L104, the reference to Fig. x is missing.
>
> Thank you for catching this. Due to a typographical error, our data distribution figure is not displayed correctly in the PDF file. We sincerely apologize for the formatting error that caused **Figure. x**. The reference in Line 104 originally pointed to a Data distribution pie chart that summarizes the number of samples curated under each evaluation perspective and the selection method (human verification vs. confidence-based filtering).
>
> To clarify, below is a summary of the dataset statistics:
>
> | Perspective          | Total Samples | Human-Selected | Confidence-Selected |
> |:----------------------:|:---------------:|:----------------:|:---------------------:|
> | Alignment            | 6260          | 2489           | 724                 |
> | Safety               | 4852          | 2271           | 574                 |
> | Quality              | 5964          | 1680           | 1121                |
> | Bias (group-wise)    | 140           | 105            | 18                  |
> | Composition          | 2829          | 1290           | 780                 |
> | Visualization        | 1600          | 797            | 692                 |
>
> We will restore the missing figure in the revised version and ensured that all references are now correctly displayed. Thank you again for pointing this out.
>
>
> We hope that our detailed explanation of the six perspectives, the clarified mathematical notation, and the correction of the missing figure have addressed your concerns. Please don’t hesitate to let us know if anything remains unclear or if further clarification would be helpful.
>
> If all your concerns have been resolved, we would sincerely appreciate it if you could consider raising your score to help us share this work with a broader community.
>
> Thank you again for your time and thoughtful feedback!
>
> Best regards,
>
> Submission #884 Authors
>
>
> **Reference**
>
> [1] Black, Kevin, et al. "Training diffusion models with reinforcement learning." arXiv preprint arXiv:2305.13301 (2023). \
> [2] Lee, Tony, et al. "Holistic evaluation of text-to-image models." Advances in Neural Information Processing Systems 36 (2023): 69981-70011. \
> [3] Wan, Yixin, et al. "Survey of bias in text-to-image generation: Definition, evaluation, and mitigation." arXiv preprint arXiv:2404.01030 (2024). \
> [4] Wang, Boxin, et al. "DecodingTrust: A Comprehensive Assessment of Trustworthiness in GPT Models." NeurIPS. 2023. \
> [5] Zhou, Kankan, Yibin LAI, and Jing Jiang. "Vlstereoset: A study of stereotypical bias in pre-trained vision-language models." Association for Computational Linguistics, 2022.

---

> > ### Comment · Reviewer_PPqG · 2025-08-01
> > **Response to the authors**
> >
> > Dear authors,
> >
> > Thank you for addressing my concerns.
> >
> >
> > I understand why the six perspectives are employed.
> > Although other perspective can be considered, the employed ones are most important perspective.
> > These are also supported by prior works.
> >
> >
> > Authors also revise the notation and typos.
> >
> >
> > Since all of my concerns are solved, I raise the score.

---

> > > ### Author Response · Authors · 2025-08-04
> > >
> > > Dear Reviewer PPqG,
> > >
> > > We are glad that we have addressed all your concerns! We deeply appreciate your positive feedback and your decision to raise your score! We sincerely thank you for the time and effort you have dedicated to reviewing our paper and helping us further improve our work. We will carefully follow your suggestions and incorporate all updates into the new version. Thank you again for your valuable support!

---

### Official Review · Reviewer_KPGU · 2025-07-05

**Ethics Flags:** Discrimination, bias, and fairness
**Rating:** 4
**Confidence:** 4

**Summary:**

This paper introduces MJ-BENCH, a comprehensive benchmark designed to evaluate multimodal reward models—particularly in the context of text-to-image generation. It evaluates a broad spectrum of models (including both CLIP-based scorers and vision-language models) across six key dimensions: alignment, safety, image quality, bias, composition, and visualization. The benchmark features fine-grained subcategories, human-validated preference data, and supports both automatic and human evaluations. Empirical results demonstrate that closed-source VLMs (e.g., GPT-4o) provide the most robust feedback across dimensions, while smaller models exhibit domain-specific strengths.

**Dataset Code Accessibility:**

Yes

**Ethical Considerations:**

Yes, there are ethics concerns that require attention by the authors

**Final Justification:**

My concerns about could MJ-BENCH be leveraged to train new reward models have been well addressed, so my final justification is **Borderline Accept**.

**Limitations Weaknesses:**

* Although the dataset was human-verified, no quantitative measure (e.g., agreement rates, annotation consistency) is reported, which makes it difficult to assess the overall quality and reproducibility of MJ-BENCH.
* The paper is somewhat lengthy and information-dense, especially in the dataset construction sections. Some details (e.g., perturbation strategies, prompt editing) could be moved to the appendix to improve readability.
* While the benchmark is well-constructed, the paper does not sufficiently explore how MJ-BENCH could be leveraged to train new reward models or improve alignment techniques beyond evaluation.

**Strengths Contributions:**

*  The finding that small CLIP-based models can outperform large VLMs on text-image alignment and image quality, while VLMs are better at assessing safety and bias, provides practical guidance for model deployment and reward tuning.
* The paper evaluates a wide variety of scoring models and vision-language models (VLMs), both open- and closed-source, under multiple settings (e.g., single- vs. multi-input, numerical vs. Likert scales). The authors also cross-validate the findings using human evaluations, which adds credibility to the conclusions.

---

> ### Author Rebuttal · Authors · 2025-07-31
>
> Dear Reviewer KPGU,
>
> Thank you for your appreciation of the value and novelty of MJ-Bench! Following your suggestions, we have revised the paper to include additional experiments and discussions, and we hope these address your concerns.
>
> > Although the dataset was human-verified, no quantitative measure (e.g., agreement rates, annotation consistency) is reported, which makes it difficult to assess the overall quality and reproducibility of MJ-BENCH.
>
> We sincerely thank the reviewer for raising this important point. We fully agree that quantitative evaluation of the annotation process is essential for ensuring the quality and reproducibility of MJ-Bench. As a matter of fact, we have already incorporated a systematic analysis of our quality control process including statistics on human agreement in Appendix B.1.
>
> For example, Table-r.1 (also Table 5 in the appendix) presents the number of validated examples at each stage of our multi-step quality control process.
>
> _Table-r.1: Statistics of dataset samples during the quality control process_
>
> |                   | Alignment | Safety | Quality | Bias (group) | Composition | Visualization |
> |-------------------|-----------|--------|---------|--------------|-------------|---------------|
> | Total             | 6260      | 4852   | 5964    | 140          | 2829        | 1600          |
> | Human Selected    | 2489      | 2271   | 1680    | 105          | 1290        | 797           |
> | Confidence Selected | 724     | 574    | 1121    | 18           | 780         | 692           |
>
> Following your insightful suggestion, we also evaluated inter-annotator agreement rates to assess annotation consistency. Specifically, each sample was verified by at least two human experts, and we counted agreement when a majority shared the same preference (e.g. 2/2, 2/3). We report the results in Table-r.2. As expected, agreement is higher on more objective perspectives (e.g. alignment, safety, bias, composition), and relatively lower on subjective aspects (e.g. quality, visualization). To ensure fairness and consistency of our evaluation, we retained only those samples where a substantial majority (≥80%) agreed, resulting in the final numbers reported above in Table-r.1.
>
> _Table-r.2: Agreement rate across different human annotators_
>
> | Perspective   | Agreement rate (%) |
> |---------------|-------------------|
> | Alignment     | 87.9              |
> | Safety        | 96.0              |
> | Quality       | 75.4              |
> | Bias          | 93.7              |
> | Composition   | 89.4              |
> | Visualization | 81.2              |
>
> To further demonstrate the quality of our dataset, we fine-tuned a text-to-image model (SD-1.5) using preference pairs from MJ-Bench, and compared its results to both the base model and a model fine-tuned with GPT-4o feedback. Due to time constraints, we only reported results on a subset of perspectives in the initial submission (Table.6 in Appendix B.1). However, Table-r.3 below now provides complete human evaluation rankings for all six perspectives.
>
> _Table-r.3: Human evaluation result on images generated from three models_
>
> | Dataset Configuration | Alignment | Safety | Quality | Bias | Composition | Visualization |
> |----------------------|-----------|--------|---------|------|-------------|---------------|
> | SD-1.5 Base          | 2.47      | 2.70   | 2.23    | 2.63 | 2.58        | 2.43          |
> | SD-1.5 + GPT-4o      | 1.95      | 1.91   | **1.87**| 2.11 | 2.11        | 1.82          |
> | SD-1.5 + MJ-Bench    | **1.58**  | **1.39**| 1.90   | **1.26**| **1.31** | **1.75**      |
>
> We hope these additional statistics and clarifications better convey our quality control procedures and address your concerns regarding the reproducibility and value of MJ-Bench.
>
>
> > The paper is somewhat lengthy and information-dense, especially in the dataset construction sections. Some details (e.g., perturbation strategies, prompt editing) could be moved to the appendix to improve readability.
>
>
> We thank the reviewer for this constructive suggestion. In response, we have moved detailed descriptions of perturbation strategies, prompt editing, and other technical procedures to the appendix to improve the readability and flow of the main text. We hope these changes can make the paper more accessible and will ensure to incorporate them in the revised version.
>
>
> > While the benchmark is well-constructed, the paper does not sufficiently explore how MJ-BENCH could be leveraged to train new reward models or improve alignment techniques beyond evaluation.
>
> We completely agree with the reviewer’s insightful point about the potential of MJ-Bench for training new reward models and advancing alignment techniques. As a matter of fact, we have already conducted such experiments and present the results in Section C.7 (and also Table-r.4 below), where we trained a MoE-based reward model using a held-out subset of the human-validated data (with no overlap with the evaluation set). Due to time constraints at submission, we report results from a preliminary model covering a subset of perspectives, and we plan to extend this to all six perspectives in the revised version.
>
> _Table-r.4: Evaluation results of our MoE-based judge model trained on a split from MJ-Bench. We compare a subset of the best-performing models from Table 2 using the remainder of the data as a test set. Best results are bolded._
>
> | Model                          | Alignment Avg w/ Tie | Alignment Avg w/o Tie | Safety Avg w/ Tie | Safety Avg w/o Tie | Quality Avg w/ Tie | Quality Avg w/o Tie | Bias ACC | Bias NDS | Bias GES |
> |--------------------------------|----------------------|-----------------------|-------------------|--------------------|---------------------|----------------------|----------|----------|----------|
> | GPT-4o              | 58.7               | 63.0                 | 43.2              | 97.3               | 93.5                | 95.2                 | 66.3     | 84.9     | 91.2    |
> | LLaMA-3.2-11B-Vision | 60.2                | 64.2                 | 38.1              | 80.0               | 68.5                | 74.3                 | 83.0     | 84.5     | 89.5     |
> | HPS-v2.1             | 42.2                | 64.3                 | 18.6              | 40.0               | 68.3                | 88.4                 | 57.4     | 74.1     | 86.6    |
> | MJ-Bench             | **71.2**            | **72.0**             | **77.0**          | 80.2               | 90.6                | 94.2                 | **86.1** | 84.7     | 90.1     |
>
> As shown in Table-r.4, the MoE-based judge model trained on MJ-Bench outperforms other models in alignment, safety, and bias (w/ tie scores), and is very close to GPT-4o on the quality subset. These results highlight both the advantages of MoE structures for multi-objective feedback and the high quality of MJ-Bench data. The findings also suggest that expanding MJ-Bench, especially for the quality subset, could further improve performance, potentially surpassing GPT-4o. We plan to continue scaling up our reward model training and evaluating it on the full MJ-Bench test set for future revisions.
>
> We hope these additional experiments and expanded discussions on the broader utility of MJ-Bench address your concerns. Please let us know if you have any further questions or would like additional clarification.
>
> If your concerns have been resolved, we would sincerely appreciate it if you would consider raising your score to help us share this work with the broader community.
>
> Thank you again for your thoughtful feedback!
>
> Best regards,
>
> Submission #884 Authors

---

> > ### Author Response · Authors · 2025-08-08
> >
> > Dear Reviewer KPGU,
> >
> > We deeply appreciate your thoughtful feedback and constructive suggestions, which have been invaluable in strengthening our work. As outlined in our previous response, we have carefully addressed your concerns and incorporated the following improvements:
> >
> > - We reported detailed quantitative measures of our human-in-the-loop quality control process, including (a) statistics for data selection at each stage, (b) inter-annotator agreement rates, and (c) end-to-end quality verification by fine-tuning SD-1.5 using our dataset. We believe these three aspects demonstrate the high quality of our data samples.
> > - We provided results on training a MoE-based reward model using a held-out subset of MJ-Bench data, which achieves SOTA performance compared to existing judge models across multiple perspectives (see Section C.7 or Table-r.4 above), highlighting the potential and broader impacts of MJ-Bench.
> >
> >
> > We wanted to kindly follow up to ensure that our responses have fully addressed your concerns. Please let us know if there are any remaining questions or if further clarification would be helpful. Thank you again for your time and for helping us further improve our work!
> >
> > Best regards,
> >
> > Submission #884 Authors

---

> > ### Comment · Reviewer_KPGU · 2025-08-08
> > **Official Comment by Reviewer KPGU**
> >
> > Thank you for the detailed response and for running the additional experiments under the tight time constraints.
> >
> > All of my concerns have been very satisfactorily addressed. Thus I maintain my score for acceptance of this work.

---

### Note · Authors · 2025-08-16

Dear Area Chairs and Reviewers,

We sincerely thank all ACs, Ethics Reviewers, and reviewers (KPGU, PPqG, eVhG, QC8n) for your insightful feedback and for engaging in constructive dialogue. We are delighted that our responses have addressed all concerns raised during the review period, and we are grateful to have received positive recommendations for acceptance from all reviewers.

Below, we summarize the discussions with the reviewers and the revisions we have made:

- **Dataset quality & reproducibility (KPGU)**: We now report detailed inter-annotator agreement (75–96 % across perspectives), multi-stage quality-control statistics, and an end-to-end SD-1.5 fine-tuning experiment that confirms the dataset’s utility.
- **Generalizability (eVhG)**: New RLAIF results on SDXL replicate the trends observed on SD-1.5, strengthening the claim that MJ-Bench is architecture-agnostic.
- **Fine-grained diagnostics (eVhG)**: We provide subcategory-level scores for Alignment (object, attribute, action, location, count) and Composition (Classical/Structural Mechanics, Optics), clarifying where each class of judges excels or fails.
- **Bias metrics (eVhG)**: We expanded the discussion of ACC, GES, NDS, illustrating how they jointly surface both overt and subtle demographic biases, and we will release per-group statistics for additional transparency.
- **Novelty & presentation (QC8n)**: We will move the head-to-head dataset comparison (ex-Table 4) into the main text and have clarified MJ-Bench’s unique scope—first benchmark to jointly cover alignment, safety, quality, bias, composition, and visualization with systematic human validation and end-to-end reward-model training.

We will also relocate the lengthy dataset-construction details to the appendix to improve readability (KPGU), correct any remaining typos, and further refine the formatting (PPqG) in response to the reviewers’ suggestions. All additional analyses will be made publicly available through an open-source release.

Thank you again for guiding us to make MJ-Bench more robust, interpretable, and useful to the community.

Best regards,

Submission #884 Authors

---

### Decision · Program_Chairs · 2025-09-18

**Decision:**

Accept (poster)

**Comment:**

The paper proposes MJ-BENCH, a comprehensive benchmark to evaluate multimodal reward models in the context of text-to-image generation. It evaluates a wide range of models (both CLIP-based scorers and vision-language models, both open-source and closed-source models) across six key dimensions: alignment, safety, image quality, bias, composition, and visualization. The benchmark employs both automatic and human evaluations. Its analyses offer practical guidance for model deployment and reward tuning.

The paper received 3 Borderline Accept and 1 Accept scores pre-rebuttal. The rebuttal addressed all reviewers' concerns, and one reviewer increased his score from Accept to Strong Accept. Overall, the reviewers agreed that the paper addressed a much-needed problem, conducted comprehensive analyses, and offered useful insights.

The Ethics Reviewers suggested including some extra discussions. However, these issues seem minor and do not affect the decision on the paper.

The ACs checked and agreed on paper acceptance. The authors should consider the comments from the Reviewers and Ethics Reviewers to revise the camera-ready version.